# Beyond Mortality: The Social and Health Impacts of COVID-19 among Older (55+) BIPOC and LGBT Respondents in a Canada-Wide Survey

**DOI:** 10.3390/healthcare11142044

**Published:** 2023-07-17

**Authors:** Robert Beringer, Brian de Vries, Paneet Gill, Gloria Gutman

**Affiliations:** 1School of Public Health and Social Policy, University of Victoria, Victoria, BC V8W 2Y2, Canada; 2Gerontology Program, San Francisco State University, San Francisco, CA 92262, USA; bdevries@sfsu.edu; 3Dalla Lana School of Public Health, University of Toronto, Toronto, ON M5S 2V8, Canada; paneet.gill@mail.utoronto.ca; 4Gerontology Research Centre, Simon Fraser University, Vancouver, BC V6B 5K3, Canada; gutman@sfu.ca

**Keywords:** health equity, intersectionality, marginalized populations, minority older adult Canadians

## Abstract

This study focused on the effects of the COVID-19 pandemic on the marginalized populations—specifically Black and Indigenous people as well as People of Color (BIPOC) compared to White older adults and LGBT individuals compared to heterosexual older adults. Data were derived from our national online survey of Canadians aged 55+, conducted from 10 August to 10 October 2020. The survey explored the influence of COVID-19 on lifestyle changes, well-being, and planning for the future. Our sample comprised 4292 respondents. We compared sets of dichotomous variables with White vs. BIPOC, LGBT vs. heterosexual, and LGBT White vs. LGBT BIPOC respondents. Significantly more BIPOC than White individuals reported changes in accessing food (44.3% vs. 33.2%) and in family income (53.9% vs. 38.9%) and fewer reported feeling accepted and happy, and more felt isolated and judged. Significantly more LGBT than heterosexual respondents reported changes in routines and in accessing social support, medical and mental health care and more feeling depressed, lonely, anxious, and sad. More LGBT–BIPOC than LGBT–White respondents reported changes in access to food (66.7 vs. 30.6, *p* < 0.001); in family income (66.7 vs. 41.5, *p* < 0.005); and in access to mental health care (38.5 vs. 24.0, *p* < 0.05). The only difference in emotional response to COVID-19 was that more BIPOC–LGBT than White–LGBT respondents reported feeling judged (25.9 vs. 14.5, *p* < 0.05). These findings reflect a complex mix of the effects of marginalization upon BIPOC and LGBT older adults, revealing both hardship and hardiness and warranting further research.

## 1. Introduction

On 11 March 2020, the World Health Organization declared COVID-19 to be a pandemic, reporting the virus had spread to over 114 countries worldwide [1]. Early on, we also learned that those who are Black, Indigenous, or People of Colour (BIPOC) had higher infection rates compared to White persons [2,3,4,5]. While mortality rates are the gravest concern, COVID-19 is also known to have caused psychological stress, anxiety, and depression [6,7,8,9,10,11] and significant disruption to daily lives. These more hidden consequences—the focus of this paper—are less understood, including how they might interact with other experiences of daily life for some older adults such as stigma and discrimination.

The US National Academies of Sciences, Engineering and Medicine point out that “health inequities… are the result of the historic and ongoing interplay of inequitable structures, policies, and norms that shape lives” [12] (p. 8). The concept of race is a social construct carrying extensive consequences for individuals who are part of a minority [13]. While “race” differs and may be considered outdated in comparison to the term “ethnicity”, it is used here as it is focused on the physical characteristics which are nevertheless found to impact the day-to-day lives of Canadians. Paramount among these consequences is racism, wherein these social groups (races) are often hierarchically compared (typically in a negative manner) to the dominant social group [14] and relatedly discrimination. In Canada, Black Canadians experience racism and discrimination across multiple domains, including education (lower levels of formal educational achievement compared to White Canadians), employment (higher unemployment rates and a more likely status among the “working poor”), housing (more live in unaffordable and substandard housing) and food insecurity (Black Canadians report moderate or severe food insecurity 2.8 times more than White Canadians) [15]. Black Canadians also report poorer health outcomes related to COVID-19 symptoms, treatment and hospitalization compared to the national average [16].

A similar pattern follows for Indigenous people in Canada, where this group ranks as Canada’s most socially disadvantaged and marginalized population [17]. There is a large wage gap when Aboriginal and non-Aboriginal earnings are compared, with Indigenous people faring poorly [18]. Unfortunately, income-related health inequities among Indigenous Canadians increased by 23% from 2001 to 2012, with Indigenous peoples subject to higher rates of chronic conditions such as obesity and diabetes as well as an increased prevalence of substance abuse, addiction, and suicide [19,20]. This culminates in a much lower life expectancy in comparison to their non-Indigenous Canadian counterparts.

In Canada, the term “visible minority” is used for data collection, operationalized (by the Employment Equity Act) as persons other than Indigenous who are members of visible minority populations such as Chinese, Black, Filipino, South Asian, Japanese, Southeast Asian, Latin American, Korean and Arab [21]. The acronym BIPOC has more recently been adopted when describing this population. With Black populations being included in this definition, we acknowledge there is an overlap in terms of data we have reviewed pertaining only to this community. Additionally, when grouping in aggregate such a diverse number of communities, there is the potential for data to regress towards the mean (or better). For example, Catalyst [22] highlighted that people of colour in Canada have higher levels of education (42% completed post-secondary education compared to 28.5% of the general population), and these levels are increasing in terms of the percentage of those in our workforce. Similarly, rates of arthritis and asthma are higher among White Canadians versus people of colour [23]. At the same time, people of colour in Canada receive less pay compared to their White counterparts and suffer emotional discomfort (expectation of difficulties based upon race) in the workplace [22]. In terms of housing, people of colour occupy substandard housing 1.8 times more often than non-visible-minority Canadians [23]. The findings described here point to the diverse experiences of BIPOC Canadian older adults across multiple domains, including both resilience and challenges.

Of course, stigma derives from many factors beyond race; the recent literature has highlighted the stigma and discrimination experienced by sexual and gender minority (SGM) persons, particularly older adults. Emlet [24], for example, notes that lesbian, gay, bisexual and transgender (LGBT) older persons face health disparities and inequities comparable to those “disadvantaged due to income, education level, and racial and ethnic background” (p. 17). These include economic and healthcare disparities, higher rates of disability, poorer physical health, poorer mental health, particularly depression, among others [25]. Emlet points to the differences within subgroups of the LGBT population, even though much of the literature has treated the LGBT population homogenously [24].

### 1.1. Intersectionality

These layered differences are particularly noteworthy, frequently described as intersectionality, referring to multiple interlocking identities [26]. Identifying as LGBT and BIPOC has the potential to magnify this effect in terms of marginalization for an individual [27,28]. Lesbians, for example, without even factoring in race, have a higher prevalence of asthma, arthritis, obesity, heavy drinking, smoking, and lower self-rated mental health in comparison to heterosexual women in Canada [23]. Abdillahi [15] has reported that the intersection of being Black with sexual orientation, gender identity, and immigration status represents a significant gap in the research literature, a gap we believe exists across the entire BIPOC spectrum.

Further, each BIPOC subgroup is subject to a number of complexities; for example, there are numerous differences among Indigenous Canadians depending on whether they are First Nations, Metis, or Inuit and whether or not they live on or off reserve [20]. This introduction is not meant to be comprehensive, but rather presents a background that sets the stage for our research questions.

### 1.2. Research Questions

Gonzales et al. [29] draw attention to important factors contributing to these inequities in the context of the pandemic: people of colour, when compared to their White peers, have fewer resources to help buffer the impacts of COVID-19. The central question of our research focuses on the potentially exacerbated effects of the pandemic on marginalized populations, specifically BIPOC individuals when compared to White older adults and LGBT individuals when compared to heterosexual older adults. We address this question through the analysis of data collected in a national survey of older Canadians aged 55+. We hypothesize that:Those who are BIPOC will report more negative outcomes across a range of COVID-19 health, emotional and social domains in comparison to those who are White.Those who are LGBT will report more negative outcomes across a range of COVID-19 health, emotional and social domains in comparison to those who are heterosexual and cisgender.

We also propose intersectional effects and hypothesize that:3.Those who are BIPOC and LGBT will report more negative outcomes in comparison to those who are White and LGBT.

## 2. Materials and Methods

This study is based upon an online survey that focused on current experiences and planning for the future during the COVID-19 pandemic. The survey, open to Canadians aged 55 and over, sought to explore pandemic-related stressors experienced by older adults, including access to health and social care. We indicated that we were seeking respondents from the general population as well as targeting responses from minority groups, including those who self-identify as LGBT as well as members of Canada’s most populous racial minorities. Respondents were recruited using Facebook advertising and direct email; we enlisted assistance with recruitment from over 85 organizations serving older adults in general and the three sub-populations of interest: LGBT, South Asian and Chinese older adults (Canada’s two largest ethnic minorities). Potential respondents were directed online to a consent page which described their rights as research participants and from which, upon indicating consent, they could access the survey. The study was approved by the Simon Fraser University’s Research Ethics Review Board.

The survey questions were mostly developed by the study authors, informed by our work as members of The Diversity Access Team (DAT), which is part of a larger pan-Canadian study focused on assessing, tailoring, implementing, and evaluating Advance Care Planning (ACP) tools aimed at minority populations. The 61-item survey, which opened on 10 August 2020 and closed on 10 October 2020, included a blend of Likert-scale items and open-ended questions, and respondents spent on average just over 13 min completing it. A report describing the recruitment method and a detailed description of respondents’ socio-demographic characteristics can be found elsewhere (available at: www.sfu.ca/lgbteol.html (accessed on 15 October 2022)).

### Measures

Data for this study come from a subset of questions in the survey that enquired about emotional reactions to the pandemic and its impact on lifestyle. We asked respondents whether, since the outbreak of COVID-19, they felt: depressed, lonely, isolated, anxious, relaxed, sad, happy, judged by others, and accepted in their community. Respondents could choose between the following: most of the time, some of the time, or seldom/never. The impact of the pandemic on lifestyle was captured utilizing seven items from the Stoddard and Kaufman [30] Coronavirus Impact Scale (CVIS). The seven areas explored in these items were changes in the following: routine; family income and employment; food access; access to family and other social support; stress and discord in the family; medical care; and mental health treatment. Four-point Likert scales were provided in response to these domains; these response scales included options identifying either the number of change areas (e.g., no change, one to three areas with examples) or with the magnitude of changes (e.g., no change, small changes to inability to meet needs) or with the severity of the stress (e.g., none to examples of physical violence). A PDF of the full survey is available at http://www.sfu.ca/lgbteol.html (accessed on 15 October 2022).

Respondents were asked to select an “ethnic or cultural background” from the following list (and/or add an identity of their own wording): White, Chinese, South Asian, Black, Filipino, Latin American, Southeast Asian, Arab, Japanese, Korean, First Nation, Inuit, Metis, and/or a background not listed. Sexual orientation was noted in responses to a question asking whether respondents considered themselves to be “heterosexual, homosexual (lesbian or gay), bisexual, don’t know, no answer”. Respondents were asked whether they “identify as transgender” as well as whether they identified as “male, female, non-binary, or an additional category not listed”.

## 3. Results

### 3.1. Analysis Strategy

For the analyses, we created dichotomous categories for all measures, for example, combining those who listed “moderate” or “severe” changes for comparison with those who listed “no” or “mild” changes (or equivalent responses) on lifestyle variables; regarding the emotion variables, we combined those who listed “some of the time” or “most of the time” (into “at least some of the time”) for comparison with those who listed “seldom or never.” Z-tests were conducted comparing these dichotomous variables with White vs. BIPOC respondents in one set of analyses, LGBT vs. heterosexual in the second, and finally White–LGBT vs. BIPOC–LGBT in the final set of analyses, corresponding to the three hypotheses. The BIPOC category comprised those who identified as other than “White”. The LGBT category comprised those who did not identify as both heterosexual and cisgender (or non-transgender in our sample); that is, this included gay, lesbian, bisexual and transgender (and other non-heterosexual) identities.

### 3.2. Analytic Sample Characteristics

The analytic sample (i.e., those for whom data on the above sexual orientation and gender identity measures were complete—the sample upon which the following analyses were computed) comprised 4292 respondents (with the exception of education and employment measures, for which missing data reduced this number). As shown in Table 1, participants were, on average, almost 67 years old, 61% were married and just under 30% lived alone, most (48%) lived in large urban centers, with educational attainment beyond high school (almost 80% had more than a high school diploma) and most (69%) were retired, though there were differences by race and sexual orientation as reported below.

As may be seen in Table 1, relative to White respondents, BIPOC respondents tended to be younger, more likely to live in a large urban center (and less likely to live in any of the other settings), more likely to have a high school education, a certificate beyond high school, or a bachelor’s degree as their highest level of education, and more likely to be working and less likely to be retired. Comparing across the two sexual orientation groups, LGBT respondents were more likely to be younger, less likely to be married and more likely to be single and live alone. LGBT respondents were also less likely to live in rural and small urban communities and more likely to live in large urban centers; they were less likely to have a high school diploma or a certificate as their highest level of education completed and more likely to have a graduate degree; LGBT persons were also less likely to be retired and were more likely to either be employed or not working. The percentages and significant levels are reported in Table 1.

### 3.3. Hypothesis Testing

Hypothesis 1 predicted greater negative access issues and emotions for BIPOC compared to White respondents. Accordingly (see Table 2), BIPOC older adults reported higher proportion of changes in food access (44.3% vs. 33.2%, *p* < 0.001) and family income (53.9% vs. 38.9%, *p* < 0.001). At the same time, BIPOC respondents reported lower rates of change in daily routines (86.7% vs. 89.9%, *p* < 05), family support (70.0% vs. 84.9%, *p* < 0.001) and access to medical care (71.9% vs. 80.1%, *p* < 0.001) than White respondents. Regarding the measures of positive and negative emotions (see Table 3), the BIPOC sub-group reported feeling less accepted (88.3% vs. 94.4%, *p* < 0.001), less happy (81.7% vs. 86.6%, *p* < 0.005), more judged (53.5% vs. 56.9%, *p* < 0.05), though less isolated (52.5% vs. 57.8%, *p* < 0.05).

Consistent with Hypothesis 2, LGBT respondents reported higher rates of changes in access to medical and mental health care than heterosexual respondents (85.1% vs. 78.7%, *p* < 0.005; 25.2% vs. 14.3%, *p* < 0.001; respectively) as well as in access to family and social support (86.9% vs. 82.8%, *p* < 0.05) and daily routines (92.4% vs. 89.3%, *p* < 0.05) (see Table 2). LGBT respondents also more commonly reported feeling depressed (58.6% vs. 50.8%, *p* < 0.005), lonely (55.9% vs. 50.3%, *p* < 0.05), anxious (58.9% vs. 53.0%, *p* < 0.05) and sad (63.4% vs. 55.9%, *p* < 0.005) for at least some of the time relative to cisgender heterosexual respondents as reported in Table 3.

Hypothesis 3 asserted the intersection of race and sexual orientation and predicted that those who are BIPOC and LGBT would report more negative outcomes than those who are White and LGBT. As may be seen in Table 4 (left columns), relative to White heterosexual respondents, BIPOC–LGBT respondents more commonly experienced changes in access to food (66.7% vs. 30.6%, *p* < 0.001); family income (66.7% vs. 41.5%, *p* < 0.005); and mental health care (38.5% vs. 24.0%, *p* < 0.05). Regarding the measures of reported emotions (see Table 3, right columns), the only significant difference was that more BIPOC-LGBT respondents reported feeling judged (25.9% vs. 14.5%, *p* < 0.05).

## 4. Discussion

To varying degrees, support was offered for each of our three hypotheses; marginalized groups reported greater stress and stress effects from the COVID-19 pandemic; overlapping spheres of marginalization—intersectionality—appeared to further exacerbate at least some of these effects. It is alarming that the effects were noted in domains representing some of the most basic life necessities. For example, it was in food access and family income where BIPOC respondents reported greater changes/reduced access compared to their White peers. This finding is similar to what was found in the *BC COVID-19 SPEAK: Your story, our future* project of May 2020, with results indicating that White people were less likely to struggle with food insecurity and had less difficulty in “making ends meet” than British Columbians on average [31]. The Impact of COVID-19 Black Canadian Perspectives also revealed similar findings, in which Black Canadians reported much worse financial impacts from COVID-19 than the average Canadian [16]. That food access and family income difficulties were magnified because of the pandemic has important policy implications.

We also found that BIPOC LGBT respondents appeared to experience the greatest negative changes to access to food and family income; two thirds of BIPOC LGBT respondents reported these changes “at least some of the time.” This diminished socioeconomic status and its impact is supported by the literature indicating that being BIPOC and LGBT has the potential for a compounded effect in terms of marginalization [27,28]. Although Canada has proactively addressed loss of income due to the pandemic through the Canada Emergency Response Benefit (https://www.canada.ca/en/services/benefits/ei/cerb-application.html (accessed on 20 November 2022)) and other benefits, our results suggest that the underlying inequities are still prominent. We would argue that research focused on current policies to support income in minority communities, including BIPOC and LGBT persons, could help us better understand and address this important issue. The lack of disaggregated data collection in some Canadian regions also hinders the efforts of public health messaging and distribution of resources to those who need it most [32], such as food supply, therefore adding to inequities faced by certain groups.

While the findings on food access and income are troubling, BIPOC respondents reported less change in their daily routines and family support compared to White respondents. These findings may be indicative of the various ways kin support is provided amongst individuals in racial and ethnic minority groups, with White persons being more involved in non-kin networks [33]. This corresponds with the findings of Hurlbert et al. [34] who suggest that minority populations have greater social network densities and a greater likelihood of living in multigenerational households [35] and thus a stronger reliance on and greater access to kin ties during times of disasters. Perhaps these factors in combination account for the reduced disruption in daily routines and family support among BIPOC respondents and the corresponding greater changes noted by heterosexual (e.g., majority) respondents.

Congruently, research has demonstrated that kin networks are more geographically dispersed among White respondents relative to those of BIPOC, owing in part to economic resources [36]. Thus, restricted access to households other than one’s own during the quarantine mandates of COVID-19 may have impacted White respondents more strongly. Along similar lines, we found that LGBT respondents also experienced greater changes to their family and social support. Such findings may reflect the greater geographic mobility of LGBT persons relative to heterosexual persons [37], further challenging their kin access during COVID-19 restrictions.

It is similarly alarming that LGBT respondents experienced more deficits in accessing mental health and medical care in comparison to heterosexual people. Previous research has found that older members of the LGBT community often delay accessing healthcare due to stigma and discrimination experienced over the life course, and, as a result, have more unmet health and social service needs compared to heterosexual people [38]. Although based on US data, Ruprecht et al. [39] also found that sexual minority participants had lower access to mental health treatment compared to heterosexual respondents during the COVID-19 pandemic. LGBT BIPOC respondents in particular reported the greatest changes to their access to mental health care. These results support other findings that the COVID-19 pandemic increased the magnitude of pre-existing disparities in access to health care for minority groups [40]. The pandemic also manifested new stigmatization opportunities that may have impacted access to healthcare for BIPOC respondents, potentially resulting from the increase in the anti-Asian sentiment [41] or the media’s framing of COVID-19 transmission in racialized communities [42]. Our finding ties into the social determinants of health model where access to health services is a social determinant [43] with the pandemic exacerbating these pre-existing differences.

Also expected were the emotion results where LGBT respondents experienced higher levels of depression, loneliness, anxiety, and sadness in comparison to heterosexual respondents. It is difficult to determine, however, the degree to which the pandemic has exacerbated these emotions; this may represent a continuation of the status quo, since it is well established that older LGBT people experience greater depression and loneliness in comparison to their heterosexual peers [44,45]. Early pandemic data revealed that a greater proportion of LGBT persons reported adhering to social distancing guidelines than non-LGBT individuals [46,47]. Increased compliance to COVID-19 mitigation measures reported by LGBT individuals may have contributed to their heightened feelings of anxiety and potentially to their sadness, loneliness and depression from the disruption and not being able to socialize.

Those who are BIPOC were disproportionately affected by the COVID-19 pandemic in terms of COVID-19-related morbidity and mortality [48,49] and the rhetoric as that noted above, which may have resulted in lower ratings of happiness and in taking more isolation measures to protect themselves, perhaps associated with greater concerns with being judged and related lower feelings of acceptance. Furthermore, disparities in internet access between BIPOC, who consistently have less access, and White adults may have reduced opportunities for maintaining social connections virtually and contributed to BIPOC feeling more isolated [50].

That White respondents felt less judged and more accepted is almost a truism—perhaps a defining characteristic of the majority in a cultural context and the inverse of that experienced by racial minority persons. The comparatively greater feelings of judgement and lesser feelings of acceptance experienced by BIPOC respondents may arise from the frequent news coverage on the higher COVID-19 incidence rates in BIPOC communities and the racialized context of the disease; it is plausible that distress would increase among this group [51]. Therefore, BIPOC respondents may have felt more scrutinized in the actions they took and felt less acceptance from others due to news reporting that attributed high COVID-19 transmission to Canadian neighborhoods with higher proportions of racial minority groups [49]. The doubly marginalized status of BIPOC LGBT respondents stands out here in their significantly greater reports of feeling judged—in many ways, the ultimate experience of stigma.

## 5. Conclusions

This study has several important implications, including the need to identify and address COVID-19-related stress in historically and currently oppressed groups to ensure that interventions are designed with an equitable lens that does not further stigmatize the groups. While some findings indicate that the racialized older LGBT adults in our sample faced hardships that were critical (e.g., related to income and food access) and potentially affected their mental well-being, we also see that these groups persevered in using the existing support to cope (e.g., social support and previous experiences). LGBT BIPOC persons are often hierarchically compared (typically in a negative manner) to the dominant social group—heterosexual White persons [14]. While not examined in our study, it is plausible that experiences with discrimination faced by these groups function as psychosocial stressors that are associated with adverse changes in health-related outcomes [45,52,53,54]. It is therefore racism and heterosexism, rather than being BIPOC or LGBT, that function as a social determinant of health [55,56]. We interpret these findings as reflecting a complex mix of the effects of marginalization (and minority stress processes [57] as experienced by LGBT persons in general) and privilege and relative deprivation (as experienced by heterosexual and LGBT White persons) along with resilience and the moderated expectations and experiences of BIPOC LGBT persons. That our findings did not fully support an additive effect in terms of marginalization for those who are both BIPOC and LGBT seen in other studies [27,28] warrants further research. As the population ages, future research should expand on efforts to prioritize and address the health, economic, and social needs of older adults who experience multiple and interrelating structures of inequality.

This study is not without its limitations. Due to a smaller sample size, non-White respondents were collapsed to form the BIPOC category, as were sexual minority respondents to form the LGBT category. We recognize there are vast differences within groups themselves that are overlooked when amalgamating such diverse individuals. Nonetheless, it is imperative to highlight the experiences of those who are under-represented in research to create insightful and equitable policies that have often been shaped to fit heteronormative, Western viewpoints. We recommend that government policymakers and service providers adopt a more inclusive lens to better understand Indigenous ways of knowing and the impact of minority stress processes in the lives of BIPOC Canadians [57,58,59]. More research is certainly needed to isolate the experiences of subgroups that make up a larger minority (e.g., transgender persons under LGBT, or South Asians under POC), especially as COVID-19 continues to ravage non-White-majority countries. Several of our measures were single-item scales and more multidimensional, better validated measures may enhance the uncovered patterns, along with multidimensional analyses to build upon this foundational study. Although we do not analyze gender or age differences in this paper, this has been of previous focus in our main sample’s report (www.sfu.ca/lgbteol.html (accessed on 15 October 2022)). In addition, our online survey was exclusively conducted in English and therefore restricts our sample to English-speaking, computer-literate older adults.

## Figures and Tables

**Table 1 healthcare-11-02044-t001:** Demographic characteristics of analytic sample by race and sexual orientation.

Variables	Total Sample	Race			Sexual Orientation		
	N = 4292 n(%)	White (n = 3790)	BIPOC (n = 502)	*p*-Value	HT (n = 3961)	LGBT (n = 331)	*p*-Value
Age (mean in years)	66.90	67.04	65.79	<0.001¶	67.03	65.28	<0.001¶
**Relationship Status (%) ◊**							
Single	356 (8.3)	303 (7.9)	53 (10.6)	ns	285 (7.2)	71 (21.5)	<0.001
Married	2619 (61.0)	2316 (61.1)	303 (60.4)	ns	2467 (62.3)	152 (45.9)	<0.001
Widowed	533 (12.4)	473 (12.5)	60 (11.9)	ns	498 (12.6)	35 (10.6)	ns
Divorced/Separated	784 (18.3)	698 (18.4)	86 (17.1)	ns	711 (17.9)	73 (22.1)	ns
**Living Arrangement (%) ◊**							
Alone	1285 (29.9)	1148 (30.1)	137 (27.3)	ns	1152 (29.1)	133 (40.2)	<0.001
**Community (%) ◊**							
Rural	575 (13.4)	525 (13.9)	50 (10)	<0.05	551 (13.9)	24 (7.3)	<0.001
Small urban	933 (21.7)	861 (22.7)	72 (14.3)	<0.001	885 (22.3)	48 (14.5)	<0.001
Medium urban	709 (16.5)	647 (17.1)	62 (12.4)	<0.01	663 (16.7)	46 (13.9)	ns
Large urban	2075 (48.3)	1757 (46.4)	318 (63.3)	<0.001	1862 (47.0)	213 (64.4)	<0.001
**Education (%) (n = 4097 owing to missing data) ◊**							
	N = 4097	n = 3611	n = 486		n = 3777	n = 320	
High School or less	845 (20.6)	726 (19.2)	119 (23.7)	<0.05	800 (21.2)	45 (14.1)	<0.005
Certificate	1434 (35.0)	1302 (34.4)	132 (26.3)	<0.001	1351 (35.8)	83 (25.9)	<0.001
Bachelor’s degree	935 (22.8)	806 (21.3)	129 (25.7)	<0.05	853 (22.6)	82 (25.6)	ns
Graduate degree	883 (21.6)	777 (20.5)	106 (21.2)	ns	773 (20.5)	110 (34.4)	<0.001
**Employment Status (%) (n = 3940 owing to missing data) ◊**							
	N = 3940	N = 3481	N = 459		n = 3650	n = 290	
Employed	938 (23.8)	823 (21.7)	115 (22.9)	ns	849 (23.3)	89 (30.7)	<0.005
Not working	285 (7.2)	214 (5.7)	71 (14.1)	<0.001	255 (6.9)	30 (10.3)	<0.05
Retired	2717 (69.0)	2444 (64.5)	273 (54.4)	<0.001	2546 (69.8)	171 (58.9)	<0.001

Note: BIPOC = Black, Indigenous, and People of Colour; HT = Heterosexual; LGBT = Lesbian, Gay, Bisexual and Transgender; Percentages are shown within columns; ¶ Independent Sample T test; ◊ Z-Test.

**Table 2 healthcare-11-02044-t002:** Percent reporting change on Coronavirus Impact Scale Items by race and sexual orientation.

	Race			Sexual Orientation		
	White (n = 3790)	BIPOC (n = 502)	z-Value *p*-Value	HT (n = 3691)	LGBT (n = 331)	z-Value *p*-Value
Routines	89.9	86.7	2.200 0.014	89.3	92.4	−1.770 0.038
Access to Food	33.2	44.3	−4.916 0.001	34.6	33.5	0.404 0.345
Access to family and social support	84.9	70.0	8.383 0.001	82.8	86.9	−1.912 0.028
Family Income	38.9	53.9	−6.429 0.001	40.4	43.6	−1.138 0.127
Access to Medical Health Care	80.1	71.9	4.249 0.001	78.7	85.1	−2.756 0.002
Access to Mental Health Care	15.2	14.5	0.411 0.341	14.3	25.2	−5.315 0.001
Family Discord	41.1	40.0	0.471 0.319	41.1	39.3	0.639 0.261

Note: BIPOC = Black, Indigenous, and People of Colour; HT = Heterosexual; LGBT = Lesbian, Gay, Bisexual and Transgender; Percentages are shown within columns.

**Table 3 healthcare-11-02044-t003:** Negative and positive emotions experienced at least some of the time by race and sexual orientation.

Variables (%)	Race			Sexual Orientation		
	White (n = 3790)	BIPOC (n = 502)	z-Value *p*-Value	HT (n = 3691)	LGBT (n = 331)	z-Value *p*-Value
Accepted	94.4	88.3	5.280 0.001	93.7	93.1	0.430 0.334
Relaxed	83.3	83.5	−0.113 0.456	83.6	81.9	0.799 0.212
Happy	86.6	81.7	2.976 0.001	86.0	86.4	−0.201 0.421
Depressed	51.6	50.5	0.210 0.417	50.8	58.6	−2.727 0.003
Lonely	51.1	48.3	1.39 0.082	50.3	55.9	−1.957 0.025
Anxious	53.4	53.5	0.042 0.484	53.0	58.9	−2.067 0.019
Sad	56.9	53.5	1.444 0.074	55.9	63.4	−2.644 0.004
Judged	17.0	29.6	−6.835 0.001	18.8	15.4	1.529 0.063
Isolated	57.8	52.5	2.255 0.01	56.9	60.4	−1.236 0.107

Note: BIPOC = Black, Indigenous, and People of Colour; HT = Heterosexual; LGBT = Lesbian, Gay, Bisexual and Transgender; Percentages are shown within columns.

**Table 4 healthcare-11-02044-t004:** Coronavirus Impact Scale and Negative and Positive Emotions by Race–Sexual Orientation intersections.

Comparing White LGBT with BIPOC LGBT Respondents							
Corona Virus Impact Scale				Negative and Positive Emotions			
Variables (%)	White–LGBT (n = 304)	BIPOC–LGBT (n = 27)	z-Value *p*-Value	Variables (%)	White–LGBT (n = 304)	BIPOC–LGBT (n = 27)	z-Value *p*-Value
Routines	92.7	88.9	0.714 0.239	Accepted	93.1	92.6	0.098 0.460
Access to Food	30.6	66.7	−3.807 0.001	Relaxed	82.6	74.1	1.099 0.136
Access to family and social support	87.0	85.2	0.265 0.394	Happy	87.2	77.8	1.367 0.085
Family Income	41.5	66.7	−2.531 0.006	Depressed	58.9	55.6	0.334 0.371
Access to Medical Health Care	84.5	92.3	−1.091 0.138	Lonely	55.9	55.6	0.030 0.488
Access to Mental Health Care	24.0	38.5	−1.664 0.048	Anxious	58.9	59.3	−0.041 0.484
Family Discord	40.2	29.6	1.080 0.140	Sad	63.2	66.7	−0.362 0.359
				Judged	14.5	25.9	−1.661 0.048
				Isolated	59.5	70.4	−1.109 0.134

Note: BIPOC = Black, Indigenous, People of Colour; HT = Heterosexual; LGBT = Lesbian, Gay, Bisexual, and Transgender; Percentages are shown within columns.

## Data Availability

Data available on request due to restrictions, e.g., privacy or ethical. The data presented in this study are available on request from the corresponding author. The data are not publicly available due to research ethics board guidelines.

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
