# Peer review of "Beyond Mortality: The Social and Health Impacts of COVID-19 among Older (55+) BIPOC and LGBT Respondents in a Canada-Wide Survey"

_healthcare, 2023, doi:10.3390/healthcare11142044_

Round 1
Reviewer 1 Report
The manuscript addresses the COVID-19-related disparities between marginalized populations (i.e., Black, Indigenous, People of Color) compared to white older adults, as well as LGBT people compared to heterosexual older adults in Canada. It is well-written and well-structured, and it was a pleasure reading it.
In the Introduction section, I would at least mention the fact that the term “race” is outdated and the preferred term today is “ethnicity.”
Line 119: since LGBT identifies both sexual orientation AND gender identity, I am wondering whether the authors should state also “cisgender” as opposed to T (transgender), besides mentioning heterosexuality as opposed to LGB.
Line 142: Gutman et al.’s citation is lacking the reference number.
Line 192: the authors state, “much previous research” without mentioning any reference. I would suggest mentioning at least one or two works in this regard.
Line 278: I would avoid using the term “race.” The human race is one and only one, there are just differences in terms of ethnicity. The same for lines 324, 335, etc.
Finally, I suggest that the authors at least mention the minority stress framework (and the gender minority stress theory) when interpreting the results.
A little typo at line 55: “people,” not “peoples.”
English is fine (only a typo detected - see comments above).
Author Response
The manuscript addresses the COVID-19-related disparities between marginalized populations (i.e., Black, Indigenous, People of Color) compared to white older adults, as well as LGBT people compared to heterosexual older adults in Canada. It is well-written and well-structured, and it was a pleasure reading it. Thank you for your kind comments.
In the Introduction section, I would at least mention the fact that the term “race” is outdated and the preferred term today is “ethnicity.” Added: While it differs and is considered outdated in comparison to the term ethnicity, it is used here as it is focused on physical characteristics which are nevertheless found to impact the day-to-day lives of Canadians
Line 119: since LGBT identifies both sexual orientation AND gender identity, I am wondering whether the authors should state also “cisgender” as opposed to T (transgender), besides mentioning heterosexuality as opposed to LGB. Many thanks for noting this - Added: and cisgender to hypothesis #2 (now line 124)
Line 143: Gutman et al.’s citation is lacking the reference number Added: (Available at: www.sfu.ca/lgbteol.html) (Now line 149)
Line 192: the authors state, “much previous research” without mentioning any reference. I would suggest mentioning at least one or two works in this regard. Thank you for your note on this point – We have now removed: consistent with much previous research - this should not have been included in the results section, previous research is now more appropriately discussed in the discussion section.
Line 278: I would avoid using the term “race.” The human race is one and only one, there are just differences in terms of ethnicity. The same for lines 324, 335, etc. We added to the introduction: While it differs and is considered outdated in comparison to the term ethnicity, it is used here as it is focused on physical characteristics which are nevertheless found to impact the day-to-day lives of Canadians.
Finally, I suggest that the authors at least mention the minority stress framework (and the gender minority stress theory) when interpreting the results. Great suggestion. We have added (line 389): We interpret these findings as reflecting a complex mix of the effects of marginalization (and minority stress processes [57] as experienced by LGBT persons in general)… With the following new reference:
[57] Meyer, I.H. Prejudice, social stress, and mental health in lesbian, gay, and bisexual populations: Conceptual issues and research evidence. Psychological Bulletin, 2003, 129(5), 674-697. doi: 10.1037/0033-2909.129.5.674
A little typo at line 55: “people,” not “peoples.” Changed to: people
Reviewer 2 Report
A very important article, showing that in crisis situations inequalities in access to health intensify for the population of minority groups. The conclusions can be used to develop plans to compensate for these inequalities, especially in crisis situations, and to develop an appropriate health management strategy for these groups.
It seems that the article should also include recommendations for solutions for the state administration and the health care segment for the future
Author Response
A very important article, showing that in crisis situations inequalities in access to health intensify for the population of minority groups. The conclusions can be used to develop plans to compensate for these inequalities, especially in crisis situations, and to develop an appropriate health management strategy for these groups. Thank you for your kind comments.
It seems that the article should also include recommendations for solutions for the state administration and the health care segment for the future Great suggestion Starting at Line 405 we have added: We recommend that government policymakers and service providers adopt a more inclusive lens to better understand Indigenous ways of knowing and the impact of minority stress processes in the lives of BIPOC Canadians [57-59].
Three new references have been added to the paper:
[57] Meyer, I.H. Prejudice, social stress, and mental health in lesbian, gay, and bisexual populations: Conceptual issues and research evidence. Psychological Bulletin, 2003, 129(5), 674-697. doi: 10.1037/0033-2909.129.5.674
[58] Fariba Kolahdooz, Forouz Nader, Kyoung J. Yi & Sangita Sharma. Understanding the social determinants of health among Indigenous Canadians: priorities for health promotion policies and actions, Global Health Action, 2015 8:1, DOI: 10.3402/gha.v8.27968
[59] Closer to home : Urban and Away-From-Home Health and Wellness Framework. (2020). First Nations Health Authority.
Reviewer 3 Report
Dear Authors,
the paper is of interest and well structured. The rationale is clear, as well as the research questions. I also found the results and the discussion interesting.
As for the methodology, I suggest the authors to briefly include how the potential respondents were selected and a synthesis of their socio-demographic characteristics, referring to other studies for a more in-depth analysis.
Furthermore, authors could better emphasize the practical implications of the study for managers and policy-makers.
Author Response
Dear Authors,
the paper is of interest and well structured. The rationale is clear, as well as the research questions. I also found the results and the discussion interesting. Thank you for your kind words.
As for the methodology, I suggest the authors to briefly include how the potential respondents were selected and a synthesis of their socio-demographic characteristics, referring to other studies for a more in-depth analysis. Great suggestion! Starting at line 134 we added: Respondents were recruited using Facebook advertising and direct email; we enlisted assistance with recruitment from over 85 organizations serving older adults in general and the three sub-populations of interest: LGBT, South Asian and Chinese older adults (Canada’s two largest ethnic minorities). Potential respondents were directed online to a consent page which described their rights as research participants and from which upon indicating consent, they could access the survey.
Furthermore, authors could better emphasize the practical implications of the study for managers and policy-makers. Very helpful – thank you! Starting at Line 405 we have added: We recommend that government policymakers and service providers adopt a more inclusive lens to better understand Indigenous ways of knowing and the impact of minority stress processes in the lives of BIPOC Canadians [57-59].
Three new references have been added to the paper:
[57] Meyer, I.H. Prejudice, social stress, and mental health in lesbian, gay, and bisexual populations: Conceptual issues and research evidence. Psychological Bulletin, 2003, 129(5), 674-697. doi: 10.1037/0033-2909.129.5.674
[58] Fariba Kolahdooz, Forouz Nader, Kyoung J. Yi & Sangita Sharma. Understanding the social determinants of health among Indigenous Canadians: priorities for health promotion policies and actions, Global Health Action, 2015 8:1, DOI: 10.3402/gha.v8.27968
[59] Closer to home : Urban and Away-From-Home Health and Wellness Framework. (2020). First Nations Health Authority.
Reviewer 4 Report
Dear authors,
I have the opportunity to review your manuscript. The subject of this study is very interesting and could reveal hidden angles of disparity. My comments are as follows:
please provide numeric data in the results section of the abstract
As the study conducted on adults +55 Y/O, it is misleading not to mention it in the title. i.e., it is important to indicate that this is a national survey on older adults.
The study measurement tool regarding emotional reactions to the pandemic and its impact on lifestyle, is not validated. Feeling depressed or lonely, or … could be different for individuals and it could be misleading when not validated.
Does the CVIS result in a final score? Or each question could be considered separately? It is indicated that impact of pandemic on lifestyle is evaluated using 7 questions of CVIS, did you validate this new set of questions?
How was the sampling? How did you disseminate the survey?
Line 128, What do you mean by “targeting responses from minority groups”?
Line 157-158, the link to the survey is not working.
The CVIS questionnaire ask participants to indicate if there was any change in their lifestyle. But it is not clear if the change was positive or negative (it seems it is almost negative). And this could lead to biased answers.
I think the subsection of “analytic sample characteristics, is not fitted to the methods section. It should be in the results section.
What do you mean by “analytic sample”? didn’t you analyze all the respondents’ data? If not, how do you select a sample from respondents?
I think if you re-design the results section in accordance to your main hypothesis, it would be more informative. For example, the first subsection could be about the whole population, the second subsection could report data comparing BIPC and white groups, the third subsection could report data comparing LGBT and HT groups, and the forth subsection could report data regarding BIPOC+LGBT and WHITE+LGBT.
Line 192, the sentence “consistent with much previous research” is not fitted to the results section.
Table 1 needs clarification. It is not obvious if the reported p-values are related to comparison between white and BIPOC or they are related to the comparison between HT and LGBT in white and BIPOC categories. So you can say that for example LGBT respondents were more likely to live alone but where is the related statistic index?
What are the categories for living arrangements?
In table 2, and table 3, add number next to percent. These two tables include many data but not well defined. I think it would be better to change each table into two separate tables.
Line 226-230, “As may be seen on Table 2, relative to White-heterosexual respondents, LGBT-BIPOC respondents more commonly experienced changes in access to food, medical and mental health care, and family income; BIPOC-heterosexual respondents reported the lowest rates of change in access to family and social support’, there is no statistics in table 2 regarding this result.
Why there is no secondary analysis? Like regression for example.
Discussion, the structure of this section is fine. But I believe that more analysis is needed, and more analysis may change the results and also the discussion and conclusion.
Author Response
Dear authors,
I have the opportunity to review your manuscript. The subject of this study is very interesting and could reveal hidden angles of disparity. Thank you for your thoughtful comments. My comments are as follows:
please provide numeric data in the results section of the abstract – Thanks you for this suggestion, we have updated the abstract accordingly.
As the study conducted on adults +55 Y/O, it is misleading not to mention it in the title. i.e., it is important to indicate that this is a national survey on older adults.
We have changed the title to: Beyond mortality: The social and health impacts of COVID-19 among Older (55+) BIPOC and LGBT respondents in a Canada-wide survey
The study measurement tool regarding emotional reactions to the pandemic and its impact on lifestyle, is not validated. Feeling depressed or lonely, or … could be different for individuals and it could be misleading when not validated.
Does the CVIS result in a final score? Or each question could be considered separately? It is indicated that impact of pandemic on lifestyle is evaluated using 7 questions of CVIS, did you validate this new set of questions? We have considered each question separately.
How was the sampling? How did you disseminate the survey? We have added at line 135: Respondents were recruited using Facebook advertising and direct email; we enlisted assistance with recruitment from over 85 organizations serving older adults in general and the three sub-populations of interest: LGBT, South Asian and Chinese older adults (Canada’s two largest ethnic minorities). Potential respondents were directed online to a consent page which described their rights as research participants and from which upon indicating consent, they could access the survey
Line 128, What do you mean by “targeting responses from minority groups”? We had hoped to be able to have a sizeable representation of Chinese and South Asian Older Adults (Canada’s two largest ethnic minorities) and other groups.
Line 157-158, the link to the survey is not working. We have changed the link to: http://www.sfu.ca/lgbteol.html (now line 149)
The CVIS questionnaire ask participants to indicate if there was any change in their lifestyle. But it is not clear if the change was positive or negative (it seems it is almost negative). And this could lead to biased answers. Please see actual survey questions below. As you will note the response categories are graded,
Family Income/Employment (please check one):
- No change.
- Small change; able to meet all needs and pay bills.
- Having to make cuts but able to meet basic needs and pay bills.
- Unable to meet basic needs and/or pay bills.
- Food Access (please check one):
- No change.
- Enough food but difficulty getting to stores and/or finding needed items.
- Occasionally without enough food and/or good quality (e.g., healthy) foods.
- Frequently without enough food and/or good quality (e.g., healthy) foods.
I think the subsection of “analytic sample characteristics, is not fitted to the methods section. It should be in the results section. Thank you for this suggestion, we have moved the sample characteristics to the beginning of the results section.
What do you mean by “analytic sample”? didn’t you analyze all the respondents’ data? If not, how do you select a sample from respondents? This section has been revised as follows:
The analytic sample (i.e., those for whom data on the above sexual orientation and gender identity measures were complete—the sample upon which the following analyses are computed) comprised 4292 respondents (with the exception of education and employment measures, for which missing data reduced this number). As shown in Table 1, participants were, on average, almost 67 years old, 61% were married and just under 30% lived alone, most (48%) lived in large urban centers, with educational attainment beyond high school (almost 80% had more than a high school diploma) and most (69%) were retired, though there were differences by race and sexual orientation, as reported below.
<insert Table 1 here>
As may be seen on Table 1 relative to white respondents, BIPOC respondents tended to be younger, more likely to live in a large urban center (and less likely to live in any of the other settings), more likely to have a high school education, a certificate beyond high school, or a bachelor’s degree as their highest level of education, and were more likely to be working and less likely to be retired. Comparing across the two sexual orientation groups, LGBT respondents were more likely to be younger, less likely to be married and more likely to be single and live alone. LGBT respondents were also less likely to live in rural and small urban communities and more likely to live in large urban centers; they were less likely to have a high school diploma or a certificate as their highest level of education completed and more likely to have a graduate degree: LGBT persons were also less likely to be retired and were more likely to either be employed or not working. The percentages and significant levels are reported on Table 1.
I think if you re-design the results section in accordance to your main hypothesis, it would be more informative. For example, the first subsection could be about the whole population, the second subsection could report data comparing BIPC and white groups, the third subsection could report data comparing LGBT and HT groups, and the forth subsection could report data regarding BIPOC+LGBT and WHITE+LGBT. Thank you for the suggestion which we have adopted and which we think has strengthened the presentation of the results.
Line 192, the sentence “consistent with much previous research” is not fitted to the results section. Removed: consistent with much previous research (this should not have been included in the results section). These issues have now been more appropriately placed in the discussion.
Table 1 needs clarification. It is not obvious if the reported p-values are related to comparison between white and BIPOC or they are related to the comparison between HT and LGBT in white and BIPOC categories. So you can say that for example LGBT respondents were more likely to live alone but where is the related statistic index? Tables 1-4 have been revised.
What are the categories for living arrangements? In our survey we asked: Do you live alone? ___yes ___no. How many people do you live with?
In table 2, and table 3, add number next to percent. These two tables include many data but not well defined. I think it would be better to change each table into two separate tables. See revised tables.
Line 226-230, “As may be seen on Table 2, relative to White-heterosexual respondents, LGBT-BIPOC respondents more commonly experienced changes in access to food, medical and mental health care, and family income; BIPOC-heterosexual respondents reported the lowest rates of change in access to family and social support’, there is no statistics in table 2 regarding this result. Statistics are now included in the revised tables.
Why there is no secondary analysis? Like regression for example. Regression analysis would be the next step building upon this foundational research.
Discussion, the structure of this section is fine. But I believe that more analysis is needed, and more analysis may change the results and also the discussion and conclusion. As you will see from the tables more analysis were undertaken but the essential results in fact changed very little.